# Monolithically-grained perovskite solar cell with Mortise-Tenon structure for charge extraction balance

Fangfang Wang [1], Mubai Li [1], Qiushuang Tian[1], Riming Sun[1], Hongzhuang Ma[1], Hongze Wang[1], Jingxi Chang[1], Zihao Li[1], Haoyu Chen[1], Jiupeng Cao[1], Aifei Wang[1], Jingjin Dong[1], You Liu [1], Jinzheng Zhao[1], Ying Chu[1], Suhao Yan[1], Zichao Wu[1], Jiaxin Liu[1], Ya Li[1], Xianglin Chen[1], Ping Gao[1], Yue Sun[1], Tingting Liu[1], Wenbo Liu[1], Renzhi Li[1], Jianpu Wang [1], Yi-bing Cheng[2], Xiaogang Liu [3], Wei Huang [1,4,5] ✉ & Tianshi Qin [1] ✉

Although the power conversion efficiency values of perovskite solar cells continue to be refreshed, it is still far from the theoretical Shockley-Queisser limit. Two major issues need to be addressed, including disorder crystallization of perovskite and unbalanced interface charge extraction, which limit further improvements in device efficiency. Herein, we develop a thermally polymerized additive as the polymer template in the perovskite film, which can form monolithic perovskite grain and a unique "Mortise-Tenon" structure after spin-coating hole-transport layer. Importantly, the suppressed non-radiative recombination and balanced interface charge extraction benefit from high-quality perovskite crystals and Mortise-Tenon structure, resulting in enhanced open-circuit voltage and fill-factor of the device. The PSCs achieve certified efficiency of 24.55% and maintain >95% initial efficiency over 1100 h in accordance with the ISOS-L-2 protocol, as well as excellent endurance according to the ISOS-D-3 accelerated aging test.

Perovskite solar cells (PSCs) have been considered the most promising emerging photovoltaic technology[1–3] due to the expressive power conversion efficiency (PCE) up to 26%[4]. Although the PCE values are approaching the efficiency of monocrystalline silicon solar cells, there are still significant gaps compared to the theoretical Shockley-Queisser limit[5]. Numerous studies have been devoted to investigating and analyzing the perovskite composition[6–8], crystallization kinetics[9–13], and film morphology[14–16] of different perovskite systems, and many successful strategies have been developed to improve the efficiency of PSCs. By analyzing most of the issues with perovskite, there are two in-

depth factor that need to be addressed that limit further improvements in device efficiency.

One of the main factors is the often-mentioned issue of perovskite defects, which can lead to non-radiative recombination and thus degrade device performance[17]. As perovskite films deposited by solution processes are typically polycrystalline[18], leading to high number of structural defects in the bulk[19,20], on the surface[21], and at the grain boundaries (GBs) of the films[22,23]. In fact, most of the defects in/on the perovskite mainly stems from disordered crystallization of perovskite films during fabrication. Many solutions have been studied to enhance

[1]Key Laboratory of Flexible Electronics (KLOFE), Institute of Advanced Materials (IAM) & School of Flexible Electronics (Future Technologies), Nanjing Tech University (NanjingTech), 30 South Puzhu Road, Nanjing 211816, China. [2]Advanced Technology for Materials Synthesis and Processing, Wuhan University of Technology, Wuhan, Hubei 430070, China. [3]Department of Chemistry, National University of Singapore, Singapore 117543, Singapore. [4]Key Laboratory for Organic Electronics & Information Displays (KLOEID) & Institute of Advanced Materials (IAM), Nanjing University of Posts and Telecommunications, Nanjing, Jiangsu 210023, China. [5]Frontiers Science Center for Flexible Electronics & Institute of Flexible Electronics (IFE), Northwestern Polytechnical University (NPU), Xi'an, Shanxi 710072, China. ✉e-mail: iamwhuang@nwpu.edu.cn; iamtsqin@njtech.edu.cn

the crystallinity and surface morphology of the active layer, such as modulating perovskite formulation, optimizing deposition techniques[24–26], additive engineering[27–30], and compensatory interface passivation[31].

Another factor is the unbalanced charge extraction between the perovskite and the charge transport layer (CTL), however, not enough attention has been paid to it yet[32–34]. As a typical sandwich-structured device, not only the photoactive perovskite layer but also the charge transport layers including electron transport layer (ETL) and hole-transport layer (HTL) have a significant impact on the performance of the device, especially on reducing open-circuit voltage ($V_{OC}$) losses and suppressing hysteresis[35,36]. $V_{OC}$ and fill factor (FF) are often weakened due to undesired carrier losses at the perovskite/CTL interface during charge extraction and transport[21]. In particular, a large number of defects are predominantly located on the upper surface and at the GBs of the perovskite films, resulting in a serious trap-assisted non-radiative recombination at the perovskite/HTL interface[37,38]. Consequently, the hole extraction efficiency at the interface is substantially lower than the electron extraction efficiency, causing the interfacial space charge to form and accumulate[39]. Most of the research focused on developing new HTLs with high hole mobility or adding interfacial layers to provide gradient energy levels[40,41], however, it is still not possible to efficiently achieve balanced carrier extraction.

Herein, we used a thermally polymerized additive *N*-vinyl-2-pyr-rolidone (NVP) as a polymer template in the perovskite film, followed by a conventional HTL/Chlorobenzene (CB) solution spin-coating

process to remove the residual miscellaneous phases and open the GBs to form monolithic perovskite grains, thereby suppressing the defect-related non-radiative recombination. Furthermore, this process results in the formation of a novel "Mortise-Tenon" (M-T) structure for perovskite/HTL composite film (Fig. 1a, b), which provides an obviously larger contact area between perovskite and HTL, thereby facilitating hole extraction to achieve balanced charge management. Based on the high-quality perovskite crystalline and the unique M-T architecture that effectively enhances $V_{OC}$ and FF, PSCs can achieve certified efficiencies of 24.55% for reverse scan and 24.25% for a forward scan. Moreover, NVP-based PSCs maintain >95% initial efficiency over 1100 h in accordance with the ISOS-L-2 protocol, as well as excellent endurance according to the ISOS-D-3 accelerated aging test.

## Results
### Crystallographic characterization of monolithic perovskite grain and Mortise-Tenon structure
We used NVP as additive in the perovskite precursor, which could straightforwardly convert into polyvinylpyrrolidone (PVP) via atom transfer radical polymerization (ATRP) during perovskite annealing step. As in traditional n-i-p PSCs, the perovskite films were then covered with Spiro-OMeTAD/CB solution. To explore the effect of the additive on the architecture of the perovskite film, we operated a scanning transmission electron microscope (STEM) on the cross-sectional perovskite film fabricated by focused-ion-beam (FIB). In the overall view of STEM (Fig. 1a), we found that the perovskite film

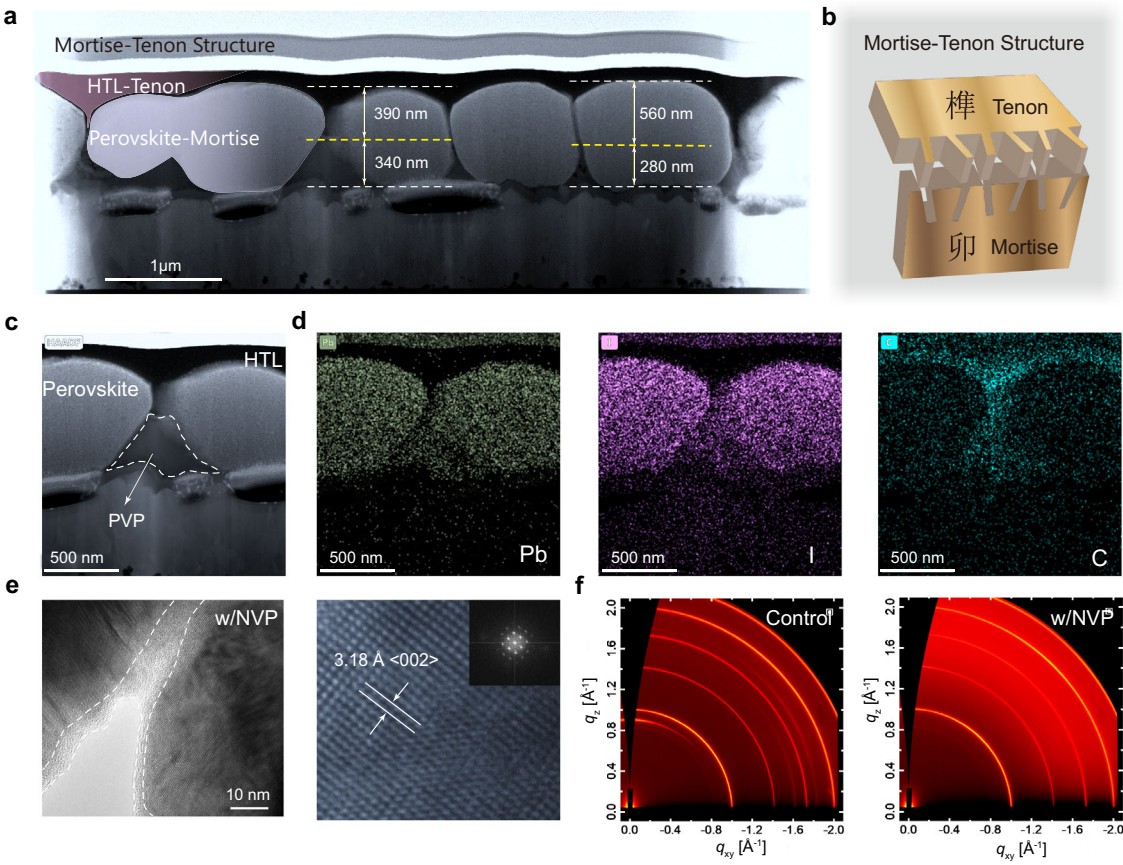

**Fig. 1 | Crystallization, architecture, and morphology of perovskite/NVP film.**
**a** Cross-sectional STEM image of an ultra-thin perovskite slice (<100 nm thickness) fabricated by FIB with an architecture (from top to bottom) of sputtered Pt/spiro-OMeTAD (HTL)/perovskite with NVP/SnO₂ (ETL)/FTO/glass. The depth of the inserted HTL depth is 1/2-2/3 of the total perovskite grain height. The yellow dashed line is the boundary between spiro-OMeTAD and dielectric barrier of PVP. **b** Diagram of Mortise-Tenon structure. **c** Enlarged images of STEM image, the

labeled triangular area may be miscellaneous phases in PVP matrix. **d** Corresponding EDS elemental mapping of Pb, I, and C from the HAADF image shown in **c**. **e** TEM image clearly shows that perovskite grains are surrounded by PVP. The crystal lattice distance of perovskite was 3.18 Å, and the SAED pattern of corresponding interdigital perovskite films as illustrated in the inset. **f** The diffraction patterns of control and perovskite/NVP films were collected by GIWAXS at a small angle.

possessed monolithic grains with the average length and width of grains exceed 1 μm. In contrast, the cross-sectional SEM images of the control film (Supplementary Fig. 1) showed small and disordered crystals. X-ray diffraction (XRD) (Supplementary Fig. 2) exhibited that the control perovskite film had a strong PbI$_2$ signal at 12.8°[42], whereas the perovskite with NVP addition (denoted as perovskite/NVP in the following discussion) featured a sharp peak of α-phase and no PbI$_2$ signal was observed. Furthermore, surprisingly, the perovskite/HTL composite film showed a M-T structure. As shown in Fig. 1b, M-T structure has been used by woodworkers for centuries, because the concave and convex parts joint each other and possess a large connection area. As shown in Fig. 1a, c, and Supplementary Fig. 3, the detailed boundary line between the inserted HTL and the perovskite was clearly observed by high-angle annular dark-field (HAADF), where the depth of the inserted HTL depth ranged from 390 to 560 nm, representing 1/2-2/3 of the total perovskite grain height. The perovskite/HTL composite films formed a unique M-T structure that could provide a larger contact area between perovskite and HTL, thereby facilitating the hole extracting. Its corresponding elemental distributions were displayed via energy dispersive X-ray spectroscopy (EDS) (Fig. 1d and Supplementary Fig. 4). Pb and I were predominantly and uniformly located in the monolithic perovskite grains, with a small distribution in the triangular area labeled in Fig. 1c. C derived from Spiro-OMeTAD was mainly found in the upper layer of perovskite and extended deep into the GBs. It was worth noting that C also dispersed in the triangular area, which might belong to PVP. This overlap in the

triangular region of Pb, I, and C element distribution might be the uncoordinated miscellaneous phases in PVP matrix, which could be considered as a dielectric zone between the HTL and ETL. We further performed high-resolution transmission electron microscopy (HR-TEM) on perovskite/NVP sample (Fig. 1e). TEM images clearly showed that the polymerized NVP/PVP gels surrounded perovskite crystal grains as amorphous phases, in which the selected area diffraction manifested an perovskite crystal lattice with <002> plane of α-phase at 3.18 Å[43]. The polymerizing could be verified by Fourier transform infrared spectroscopy (FTIR) (Supplementary Fig. 5), in which C=C stretching (1623 cm$^{-1}$) disappeared after heating at 100 °C for 1 h[44]. The grazing-incidence wide-angle x-ray scattering (GIWAXS) was further measured (Fig. 1f and Supplementary Fig. 6). The neat perovskite sample demonstrated PbI$_2$ peak at $q_z = 0.90$ Å$^{-1}$, and δ-phase rings at $q_{xy} = 1.62$ and 1.83 Å$^{-1}$, whereas perovskite/NVP film exhibited a pure α-phase without these miscellaneous signals[13]. Additionally, the diffraction halo around $q_z = 1.2 - 2.0$ Å$^{-1}$ was attributed to PVP in the perovskite films.

## The formation analysis of Mortise-Tenon structure

The above crystallographic analysis allows us to propose the main processes involved in the formation of monolithic perovskite grains and M-T structures of the perovskite/NVP film, as shown in the schematic diagram in Fig. 2a. As NVP exhibited excellent solubility for PbI$_2$ and FAI, and miscibility with DMSO, as shown in Fig. 2b, the solution of perovskite precursors with NVP/or DMSO was transparent and could

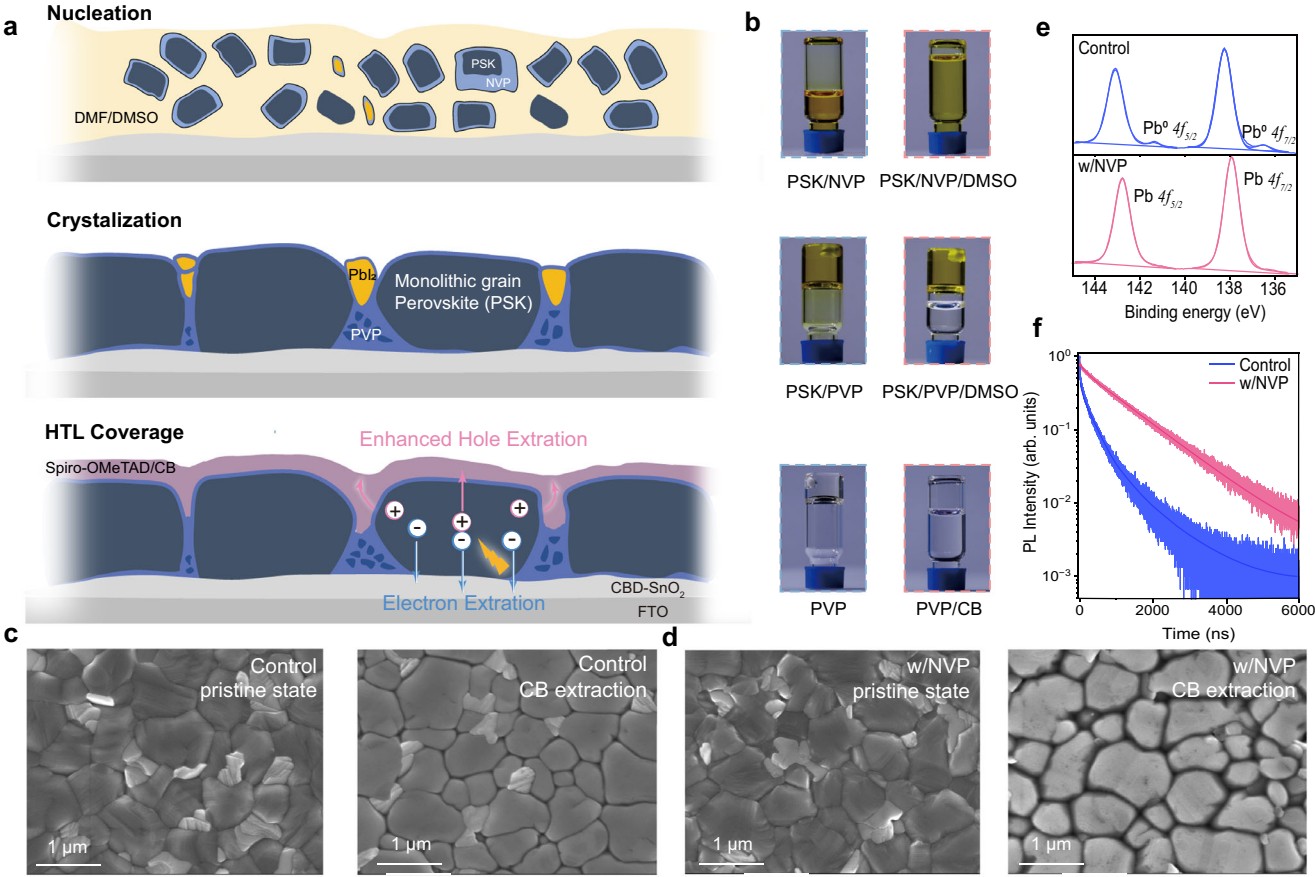

**Fig. 2 | Formation mechanism of Mortise-Tenon (M-T) structure. a** Schematic illustration of the formation mechanism of M-T structure of perovskite/NVP and upper HTL. **b** Perovskite precursor is soluble in pure NVP and is miscible with DMSO (top); perovskite precursor in NVP is polymerized to PVP after heating at 100 °C for 20 min and the solidified product is not dissolved in DMSO (middle); PVP can be dissolved in CB (bottom). SEM images of **c**, the control perovskite films and

**d**, perovskite/NVP film before and after CB extraction. **e** Pb 4*f* and I 3*d* XPS spectra of the control and perovskite/NVP film, respectively. **f** Time-resolved photoluminescence decay curves (excitation: 520 nm, 2.26 nJ cm$^{-2}$, 0.1 MHz). Solid lines were fitted from the generic kinetic model to obtain the trap density of perovskite films.

be coated uniformly. After spin-coating process, NVP could still surround the perovskite crystal seeds at the initial nucleation step owing to strong interaction between NVP and perovskite precursors (PbI$_2$ and FA$^+$) proved by $^1$H NMR, IR spectra, and a solubility experiment as shown in Supplementary Fig. 5 and Supplementary Discussion 1. During annealing process, NVP gradually polymerized into PVP and still surrounded the perovskite crystal grains, and some uncoordinated miscellaneous phases were also solidified at the GBs. As PVP is insoluble in DMSO, but soluble in CB (Fig. 2b). Therefore, after spin-coating the HTL/CB solution, CB removed the residual miscellaneous phases and opened the GBs to form monolithic perovskite grains, while remaining some PVP undissolved on the bottom as the dielectric layer. At the same time, HTL layer was covered on the perovskite films forming the unique M-T structures, which provided an obviously larger contact area between perovskite and HTL.

To verify our speculation, the scanning electron microscopy (SEM) images of control perovskite film and perovskite/NVP film were measured before and after CB extraction. HTL/CB solution was replaced with pure CB solvent, as solutions containing solute would affect the resolution of SEM surface tests. It could be observed that the miscellaneous phases (irregular bright domains) were invariably existed at perovskite grain boundaries before and after CB extraction; besides, pinholes and cracks were formed after CB extraction (Fig. 2c). For the perovskite/NVP film (Fig. 2d), the PVP and the miscellaneous phases on the perovskite and in the boundaries were effectively washed away after CB extraction, leaving distinct boundary gaps between perovskite grains, and eventually forming monolithic perovskite grains. Small molecular N-methyl-2-pyrrolidone (NMP) and PVP polymer were used as additives in perovskite films. As shown in Supplementary Fig. 7, the unpolymerized NMP additives showed a similar morphology as the control perovskite film. In contrast, the PVP polymer additive showed very small and disordered crystallization of perovskite due to rapid precipitation during annealing. Both NMP and PVP could not form the same morphological structure as NVP after CB extraction.

We further probed the chemical structure on the surface of the CB-extracting perovskite films by X-ray photoelectron spectroscopy (XPS) (Fig. 2e). The main peaks of Pb $4f_{5/2}$ and Pb $4f_{7/2}$ of perovskite/NVP film shifted by 0.54 eV towards lower binding energy compared to the control sample. In addition, after CB extracting, the Pb$^0$ peaks, which was considered to be the origin of deep defect energy level, was clearly observed in the control perovskite film at binding energy of 141.35 and 136.52 eV, whereas they completely disappeared in the perovskite/NVP films. This indicated that there was still residual PVP covered the perovskite film or in the GBs, and the lone pair electrons of the carbonyl group on PVP could effectively coordinate with Pb ions[45]. Similarly, as shown in Supplementary Fig. 8, a shift to lower binding energy was also observed in the I $3d$ spectra. The control film exhibited a C-C=O signal related to oxygen/moisture in perovskite at 288.25 eV[21], which was eliminated by adding NVP, in meanwhile, both new C=O signal at 287.63 eV and new C−N signal at 285.53 eV were detected due to carbonyl and pyrrole units on NVP molecule. In addition, perovskite films with different molar ratios of NVP addition from 15% to 60% were also measured by XPS as shown in Supplementary Fig. 8. Further shift to lower binding energy indicated that the passivation effect improved with increasing content of NVP. Furthermore, as shown in Fig. 2f, perovskite/NVP film exhibited a longer perovskite lifetime (1138.42 ns) and lower trap density ($1.58 \times 10^{15}$ cm$^{-3}$) compared to the control film (375.69 ns and $8.11 \times 10^{15}$ cm$^{-3}$) measured by time-correlated single photon counting (TCSPC) and fitted by a generic kinetic model[46,47]. Supplementary Fig. 9 showed that with increasing the excitation density, the PL quantum efficiencies (PLQEs) gradually reached a maximum value owing to the filling of defects. Perovskite/NVP film showed a higher PLQE values with a maximum of 10.20% compared to the control film (8.38%). TCSPC and PLQE measurement confirmed

that perovskite/NVP film with the monolithic grain and passivation effect synergistically suppressed the defect-related non-radiative recombination.

## Balanced charge extraction and increased built-in electric field

Atomic force microscopy (AFM, Fig. 3a) and PeakForce tunneling atomic force microscopy (PF-TUNA, Fig. 3b) images were measured under a bias voltage of 5 V to provide synchronous morphology roughness and spatially resolved electronic properties of control and perovskite/NVP films after CB extracting. The AFM topography of perovskite/NVP film depicted a higher altitude intercept (486 nm) than the control counterpart (130 nm) (Fig. 3a) owing to the M-T structure. The PF-TUNA image of perovskite/NVP film was brighter than that of the control sample, indicating that more conducting current flow through the perovskite layer in the former than in the latter, which was generally attributed to the increased conductivity and thus enhanced the charge transport[48]. The improved charge transport in overall perovskite/NVP films was ascribed to the high-quality of monolithic perovskite grains. Furthermore, the GBs of perovskite/NVP film showed brighter contrast in Fig. 3b, indicating that they carried current more efficiently. This was probably due to the thinner thickness of the film at GBs. Therefore, in this case, GBs might contribute to improving PV performance because it facilitates charge transport rather than acting as a recombination center as it usually does[22]. To better understand the charge extraction enhancement induced by M-T structure, device simulations were carried out using the commercial software Silvaco. A device with a perovskite grain size of 780 nm was designed for simulation (Supplementary Fig. 10, Supplementary Tables 1 and 2). GB grooves were assumed to be 390 nm and 560 nm deep, which was consistent with the STEM measurements (Fig. 1a). To simulate the effect of HTL extraction ability in PSCs, a 100 nm Spiro-OMeTAD was added onto the surface of perovskite film. High hole currents conducted from the GBs in the perovskite film with M-T structure could be clearly observed, which also confirmed the PF-TUNA measurement. To further verify the effect of M-T structure on the charge extraction of the perovskite film, we investigated the time-correlated single photon counting (TCSPC) characterizations on both perovskite/NVP and the control samples, with glass/FTO/ETL/PSK/HTL configurations (Fig. 3c, d and Supplementary Table 3). For the HTL side incidence test (Fig. 3c), the perovskite/NVP sample presented almost 15 times faster photo-induced luminescence lifetime (1.51 ns) than the control counterpart (22.42 ns). To further verify the origin of the enhancement of the hole extraction, ultraviolet photoelectron spectroscopy (UPS) was carried out to detect the interfacial energy level structure of the perovskite films. As shown in Supplementary Fig. 11, the control and NVP-based perovskite film exhibited comparable valence band maximum. Therefore, the effective enhancement on hole extraction came mainly from the M-T structural contact between perovskite and HTL. On the other hand, the ETL side incidence test (Fig. 3d) demonstrated similar lifetime for both interdigital (1.23 ns) and planar (1.37 ns) samples. The charge extraction balance beneficial from M-T architecture has the potential to achieve high-performance PSCs, particularly in terms of enhancing $V_{OC}$ and FF.

Surface contact potential difference (CPD) was carried out by Kelvin probe force microscopy (KPFM) with a schematic illustration in Fig. 3e, f and Supplementary Figs. 12 and 13. The height sensor diagrams (Fig. 3e) confirmed that both perovskite/NVP (667 nm) and control perovskite (630 nm) films were similar in thickness to each other. The corresponding surface CPD (Fig. 3f, g) across the perovskite/SnO$_2$ interface exhibited that perovskite/NVP film (288 mV) had a bigger CPD value than that of the control film (167 mV), respectively. In addition, Mott-Schottky measurements on entire devices consisting of glass/FTO/SnO$_2$/perovskite/spiro-OMeTAD/Au were carried out to investigate this effect on the built-in electric field ($V_{bi}$) (Fig. 3h), in which NVP-based PSC represented a higher $V_{bi}$ value

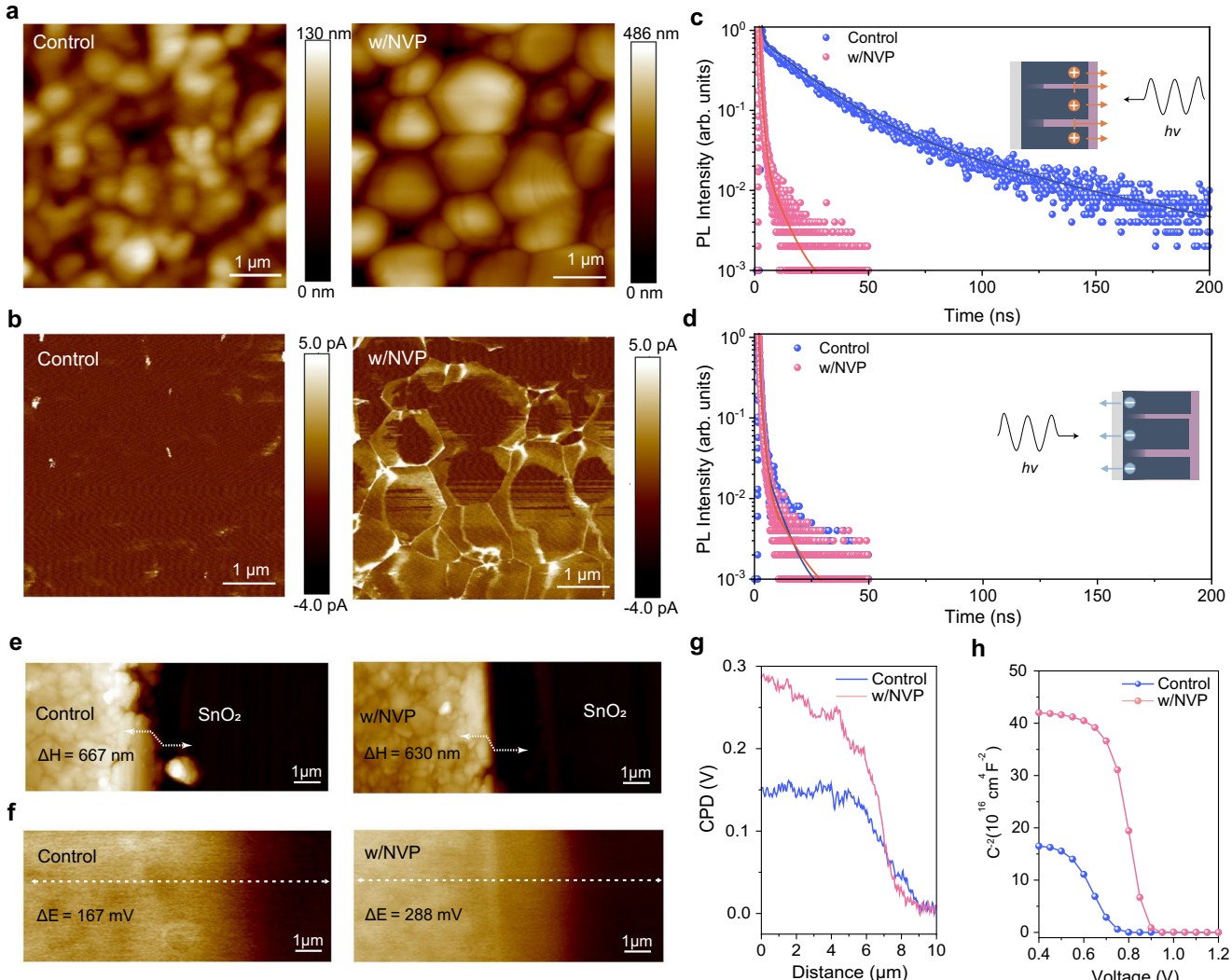

**Fig. 3 | Optoelectronic properties of the control and perovskite/NVP films.**
**a** Height images of perovskite films (5 × 5 μm). **b** PF-TUNA images of perovskite films (5 × 5 μm). TCSPC spectra of perovskite films were excited with laser (excitation: 520 nm, 55 nJ cm⁻², 1 MHz) **c** from the HTL side and **d**, from the ETL side.
**e** Surface height images and **f**, KPFM images of the interface between the perovskite and SnO₂ layers. **g** Corresponding values of surface contact potential difference (CPD). **h** Plots of C⁻² versus applied voltage by Mott–Schottky analysis in control and NVP-based perovskite solar cells.

(0.89 V) than the control counterpart (0.75 V). The space charge depletion width (W) could be calculated from a plot of $C^{-2}$ versus V. The NVP-based PSC demonstrated more than doubled W value (55.1 nm) than control PSC (24.1 nm), owing to the larger expanded space charge depletion region of the NVP-perovskite. Enhanced $V_{bi}$ and W could well facilitate charge separation and prevent carrier recombination, which was expected to realize high-performance photovoltaics, especially on improving $V_{OC}$[49].

**Photovoltaic performance and stability of PSCs**
We further explored the photovoltaic performance of PSCs with addition of NVP. PSCs were fabricated with conventional structures of glass/FTO/SnO₂/perovskite(FAPbI₃)₀.₉₅(MAPbBr₃)₀.₀₅/spiro-OMeTAD/ Au. Different molar ratios of NVP (15%, 30%, 60%, and 100%) have been used in the perovskite to obtain optimized PSCs, the current density-voltage (J–V) characteristics of PSCs were recorded under AM 1.5 G simulated solar illumination of 100 mW cm⁻² as shown Supplementary Fig. 14 and Supplementary Table 4. Among different NVP addition ratios, PSCs with 30 mol% NVP exhibited the highest device performance with the longest perovskite lifetime measured by TCSPC (Supplementary Fig. 15 and Supplementary Table 5). SEM images in Supplementary Fig. 16 clearly showed that further increasing the

proportion of NVP to 60% and 100%, large amount of NVP addition probably affected the crystallization of perovskite, and thus decreased the device efficiency. As shown in Fig. 4a, the control device had a maximum power conversion efficiency PCE of 22.91% with a $V_{OC}$ of 1.151 V, short circuit current density (Jsc) of 25.21 mA cm⁻²and a FF of 78.94%. The champion PSCs with 30 mol% NVP showed a maximum PCE of 24.69% with a $V_{OC}$ of 1.195 V, Jsc of 25.77 mA cm⁻²and a FF of 80.19%. To verify the PCE, our best NVP-based device was certificated by an independent certification laboratory (PWQC, China). This device was tested using a shadow mask with a certified size of 8.925 mm² (Supplementary Fig. 17). Supplementary Fig. 18 represented a certified PCE of 24.25% with $V_{OC}$ of 1.188 V, $J_{SC}$ of 25.41 mA cm⁻² and FF of 80.35% at foward scan (FS), and the reverse scan (RS) shows a certified PCE of 24.55%, with $V_{OC}$ of 1.187 V, $J_{SC}$ of 25.66 mA cm⁻² and FF of 80.64%. The negligible hysteresis result both in-house and certificate authority confirmed the M-T architecture of the PSCs in balancing hole and electron extraction in PSCs. It was worth noting that the NVP-based PSC exhibited excellet $V_{OC}$ and FF than the control device, which was attributed to the better crystallization of perovskite and balanced charge extraction of the M-T structural device. Electrochemical impedance spectra (EIS) in Supplementary Fig. 19 and Supplementary Table 6 showed that the NVP-based PSCs had a larger recombination

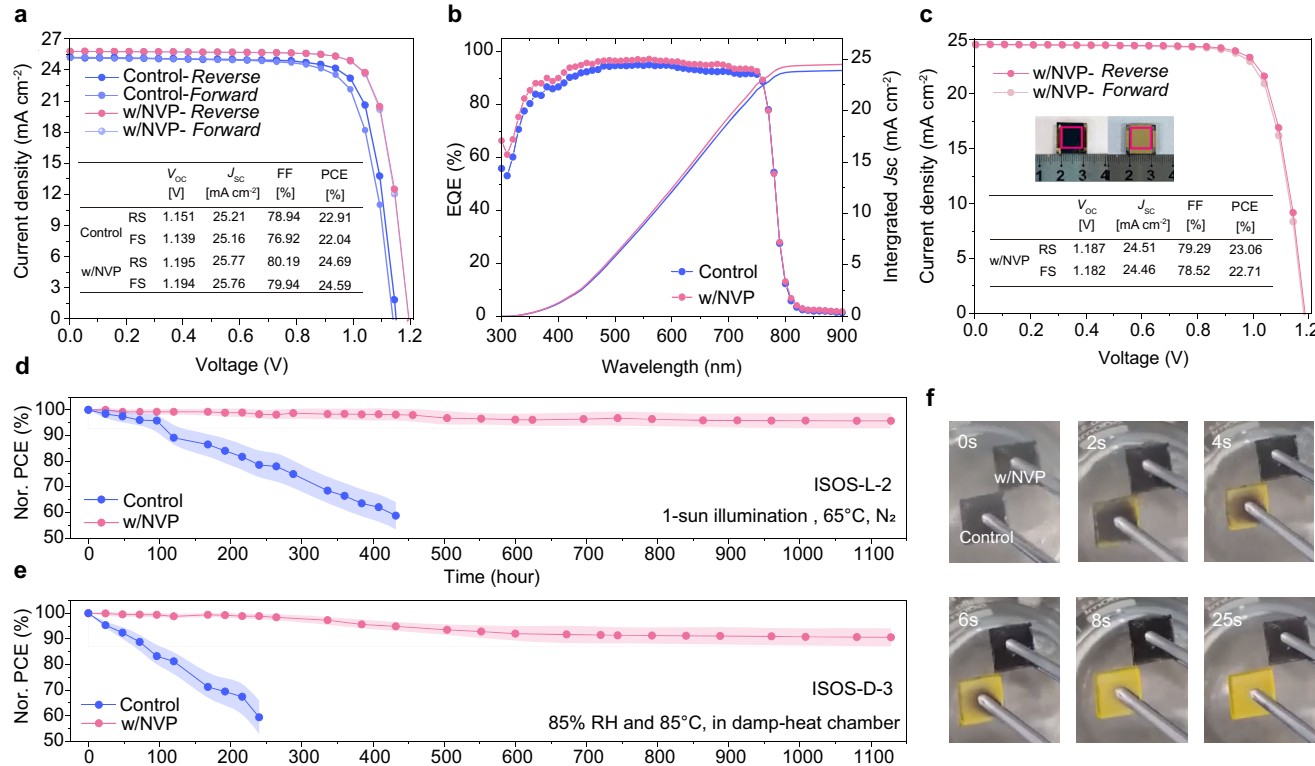

**Fig. 4 | Device performance and stability. a** $J–V$ curves and photovoltaic parameters of champion devices of the control and NVP-based PSCs measured in-house. **b** Representative EQEs and integrated $J_{SC}$ values of the control and NVP-based PSCs. **c** $J–V$ curves of a larger-area NVP-based PSC. The active area is 1.0 cm², defined by a mask aperture. The inset is a photograph of the device with dotted outlines representing the active areas. **d** Normalized PCE of encapsulated PSCs according to the ISOS-L-2 protocol (1-sun illumination and 65 °C, in N₂ atmosphere). **e** Normalized PCE of encapsulated PSCs according to the ISOS-D-3 protocol (85% RH and 85 °C, in a damp-heat chamber). All error bars represent the standard deviation of six devices. **f** Screenshots of unencapsulated control and perovskite/NVP films in water steam test (100% RH and 100 °C) captured from Movie S1.

resistance ($R_{rec}$) than the control device, which stems from the reduced non-radiative Recombination[50]. We also simulated the photovoltaic performance of the M-T structured PSC and the control device. We found that the simulated performance was very similar to our experimental results (Supplementary Table 1) with significant improvements in the $V_{OC}$ and FF of the devices. The monochromatic incident photon-to-electron conversion efficiency (IPCE) spectra showed that the integrated $J_{SC}$ values (<5% deviation) matched the $J–V$ measured data (Fig. 4b). Moreover, the NVP-based PSC exhibited steady power outputs at maximum power point (Supplementary Fig. 20) with an average PCE of 23.7% compared to their control counterparts of 22.3% (Supplementary Fig. 21). The larger PSC (1.0 cm²) showed PCEs up to 23.06% (RS) and 22.71% (FS), indicating that this M-T architecture of PSC is scalable (Fig. 4c).

By using poly[bis(4-phenyl)(2,4,6-trimethylphenyl)amine] (PTAA) instead of Spiro-OMeTAD as the hole transporting materials[21,51-55], the NVP-based PSC exhibited an excellent light exposure stability with <5% PCE loss after 1100 h at 1-sun illumination at 65 °C in N₂ atmosphere (ISOS-L-2 protocol) (Fig. 4d)[56]. In contrast, the control PSC started to decompose dramatically until the PCE loss reached 40% after 400 h. Furthermore, we also conducted the accelerated damp-heat tests of encapsulated PSCs subjected to 85% relative humidity (RH) and 85 °C, in damp-heat chamber (ISOS-D-3 protocol)[56] (Fig. 4e), where the encapsulated NVP-based PSC kept >90% initial efficiency over 1100 h. This superior stability might be attribute to the monolithic perovskite/NVP grains covered by polymerized network of PVP, which also confirmed by XPS measurement discussed above. To evaluate whether the polymer network would suppress lead leakage from the device, unencapsulated PSCs with different ratio of NVP were immersed into

deionized water for 1 hour. The lead concentration of the solution was analyzed via atomic absorption spectrometer (AAS) as shown in Supplementary Fig. 22. The Pb concentration of the control PSC was 4.86 ppm, and as the molar ratio of NVP increased from 15% to 100%, the Pb concentration decreased to 1.85 ppm, which was one-fourth of that in the control counterpart. Furthermore, owing to the excellent solubility of NVP for perovskite precursor, unencapsulated PSCs used pure NVP as the only solvent to fabricate perovskite film. The unencapsulated PSC demonstrated PCE of 17.24% and maintained >80% of its initial PCE over 600 minutes, and the control device degraded immediately (Supplementary Fig. 23). Water-vapor test provided a better visualization of the role of perovskite with NVP addition in retarding perovskite decomposition. We fumigated the unencapsulated control and perovskite/pure NVP films in hot water vapor and could see that the control films rapidly turned yellow while the NVP films remained in the black phase (Fig. 4f and Supplementary Movie 1).

## Discussion

Herein, we demonstrate an effective strategy of using a thermal polymerized NVP-perovskite/HTL composite film to form the monolithic perovskite grains and a unique M-T structure, which facilitate improved $V_{OC}$ and FF of PSCs due to the suppressed non-radiative recombination and balanced charge extraction. The corresponding PSCs exhibited excellent certified PCE up 24.55% (RS) and 24.25% (FS). The negligible hysteresis is attributed to the balance of hole and electron extraction in the PSC due to the M-T structure. Furthermore, NVP-based PSCs with polymerized network exhibited excellent illumination, moisture, and thermal stability in accordance with the ISOS-L-2 and ISOS-D-3 protocol. Our work highlights the role of increasing

the contact area of perovskite layer and HTL and proposes a unique strategy to realize interfacial charge-extracting balance in PSCs, which may be an important approach to achieve efficient device in the future.

## Methods

### Materials

Formamidinium iodide (FAI, ≥99.5%) and methylammonium bromide (MABr, ≥99.5%) were purchased from Hangzhou Perovs Optoelectronic Technology Corp (China). Methylammonium chloride (MACl, ≥99.5%), lead bromide ($PbBr_2$, 99.99%), 2,2',7,7'-tetrakis (N,N-di-pmethoxyphenylamine)–9,9'-spirobifluorene (spiro-OMeTAD, 99.5%), lithium bis(trifluoromethanesulfonyl)imide salt (Li-TFSI, ≥99%), FK209-Co(III)-TFSI (≥99%) were purchased from Xi'an Polymer Light Technology Corp (China). Lead iodide ($PbI_2$ 99.99%), N-vinyl-2-pyrrolidinone (NVP, ≥99%), azodiisobutyronitrile (AIBN, ≥98%), N,N-dimethylformamide (DMF, >99.5%), dimethyl sulfoxide (DMSO, >99.0%), chlorobenzene (CB, >98.0%), ethyl acetate (EA, >99.5%), isopropanol (IPA, >99.5%), acetonitrile (ACN, >99.5%), urea (>99.0%), and 4-tert-butyl-pyridine (TBP, >96.0%) were purchased from TCI Shanghai (China). Stannous chloride $SnCl_2 \cdot 2H_2O$ (99.99%), thioglycolic acid (TGA, 98%), and urea (≥99.5%) were purchased from Sigma-Aldrich (USA). All materials were used as received without further modification.

### Instruments

Fourier transform infrared spectroscopy (FTIR) was performed using a Scientific Nicolet iS50 spectrometer (Thermo, America). Scanning electron microscopy (SEM) was performed using JSM-7800F (JEOL, Japan). Transmission electron microscopy (TEM) was performed using JEM-1400PLUS (JEOL, Japan). GIWAXS experiments were performed on SAXS/WAXS beamline at the Australian Synchrotron. X-ray photoelectron spectroscopy (XPS) measurement was carried out on a Thermo-Fisher ESCALAB 250Xi system with a monochromatized Al Kα (for XPS mode) under a pressure of $5.0 \times 10^{-7}$ Pa. Focused-ion-beam (FIB) was performed on Helios G4 CX (FEI, USA). Scanning transmission electron microscope (STEM) was performed on Themis Z (FEI, USA). The time-resolved PL measurements were performed by a combination of a TimeHarp 260 PICO module (PicoQuant), aiHR320 spectrometer (Horiba), and COUNT-100T-FC single photon counting modules (Laser Components GmbH). PeakForce Tunneling AFM (PF-TUNA) and Kelvin probe force microscopy (KPFM) measurements were operated with the probe model SCM-PIT-V2 (material: 0.01–0.025 Ohm·cm Antimony (n) doped Si, cantilever: $T = 2.8 \, \mu m$, $L = 225 \, \mu m$, $W = 35 \, \mu m$, $f_c = 75 \, kHz$, $k = 3 \, N/m$, coating: front: conductive PtIr, back: reflective PtIr) with the Dimension FastScan AFM system, Bruker Corporation. Mott-Schottky measurement was recorded using CHI760E electrochemical workstation (CH Instruments Ins, USA). Current density-voltage (J–V) curves in-house were measured using a class 3 A solar simulator (XES-40S3, SAN-EI) under AM 1.5 G standard light equipped with a Keithley 2400 source meter. The incident photon-to-electron conversion efficiency (IPCE) measurements were carried out by a QE-R-900AD system (Nanjing Ouyi Optoelectronics Technology). The standard silicon solar cell (QE-B1) calibrated by NIM was used to calibrate the light intensity to AM 1.5 G irradiance ($100 \, mW/cm^2$). The device was measured and certified at the Quality Testing Center for Photovoltaic and Wind Power Systems of the Chinese Academy of Sciences (test report No. PWQC-WT-P21110821-1R). This certified device was tested with a shadow mask with a certified size of $8.925 \, mm^2$ provided by the National Institute of Metrology, China (test report No. CDjc2021-15963).

### Preparation of solubility measurement

To measure the solubility of NVP in the perovskite precursor, 1.474 M $(FAPbI_3)_{0.95}(MAPbBr_3)_{0.05}$ precursor was dissolved in 1 mL of pristine NVP solvent (0.3 wt.% AIBN), then heated to 60 °C and stirred for 2 h in a nitrogen-filled glovebox. The perovskite precursor in polymerizing NVP showed a viscous fluidity and adhered to the vial wall. Upon

stirring at 60 °C for another 2 h in the glovebox, the perovskite precursor in polymerized NVP became an immobilizing gel. For solubility and miscibility tests, all samples were dissolved in 1 mL of DMSO each. For measuring the polymerized NVP solubility in CB, 1 mL of neat NVP solvent each was heated at 60 °C for 4 h to avoid precipitation of the perovskite precursor in CB anti-solvent. Afterward, the polymerized NVP was completely dissolved in 1 mL of CB.

### Chemical bath deposition of $SnO_2$ layer [33]

The FTO glass was cleaned ultrasonically for 20 min with detergent, pure water, and ethanol, respectively. Then they were dried with a stream of dry nitrogen, followed by treatment with UVO for 15 min. The compact $SnO_2$ film was achieved by chemical bath deposition (CBD). The CBD solution was prepared by mixing 1.25 g of urea, 1.25 mL of HCl, 25 μL of TGA, and 275 mg of $SnCl_2 \cdot 2H_2O$ per 100 mL of ice deionized (DI) water to form a 0.012 M solution. The cleaned FTO glass was soaked in the diluted $SnCl_2 \cdot 2H_2O$ solution (0.002 M) for 2 h at 90 °C and cleaned via sonication with DI water and IPA for 5 min each. It was then annealed at 170 °C for 1 h, followed by spin-coating with 20 mM KCl in DI water at 3000 rpm for 30 s (2000 rpm ramp) and annealing at 100 °C for 10 min.

### Preparation of the perovskite layer

The perovskite precursor of $(FAPbI_3)_{0.95}(MAPbBr_3)_{0.05}$ was prepared by dissolving 240.76 mg FAI, 706.9 mg $PbI_2$, and 33.76 mg MACl, 8.21 mg MABr, 27.05 mg $PbBr_2$ salts in DMF/DMSO (8:1 v/v) mixed solvent in 1.5 M concentration. For the NVP additive system, the molar ratios of perovskite to NVP (with 0.3 wt.% AIBN) were 15%, 30%, 60%, and 100%. For typical measurements, 30 mol% was adopted. For the pure NVP system, pure NVP was used as the only solvent to dissolve the perovskite precursors. The perovskite solution was deposited on CBD-$SnO_2$/FTO by two consecutive spin-coating steps of 1000 rpm for 10 s and 5000 rpm for 30 s, respectively. During the second spin-coating step (5000 rpm), 100 μL of EA was quickly poured onto the substrate after 20 s. The films were then annealed at 100 °C for 1 h.

### Preparation of the hole-transport layer

The hole-transport layer was prepared by dissolving 30 μL of TBP, 18 μL of Li-TFSI solution (520 mg Li-TFSI in 1.0 mL acetonitrile), 29 μL of FK209-Co (III)-TFSI solution (300 mg FK209-Co (III)-TFSI in 1.0 mL acetonitrile), and 73 mg of spiro-OMeTAD in 1.0 mL CB. The hole-transport layer was deposited on perovskite films by spin-coating a spiro-OMeTAD solution at 3000 rpm for 30 s. For the device thermal stability test, PTAA doped with 4-Isopropyl-4'-methyldiphenyliodonium Tetrakis (pentafluorophenyl) borate (TPFB) was used to replace spiro-OMeTAD as the hole-transport layer. The concentration of PTAA was 30 mg $mL^{-1}$ and the weight ratio of PTAA/TPFB was 10:1. The PTAA was deposited on top of the perovskite layer at a spin rate of 3000 rpm. for 30 s.

### Sample preparation for SEM, XPS, PF-TUNA, KPFM, and GIWAXS measurement

On the FTO/$SnO_2$ substrate, 20 μL of perovskite precursor solutions (with/without NVP) were spin-coated respectively and annealed at 100 °C for 1 h. Afterward, 100 μL of neat CB solvent was dropped onto the annealed film, followed by spin-coating at 3000 rpm for 30 s. These CB-extracted perovskite films were then used for measurements.

### Sample preparation for TEM

On the FTO/$SnO_2$ substrate, 20 μL the perovskite precursor was spin-coated and annealed at 100 °C for 1 h. the perovskite powder sample was scraped from the perovskite film and dispersed in n-hexane. Then directly used for TEM measurement.

**Sample preparation for cross-sectional STEM and TCSPC measurement**

On the FTO/SnO$_2$ substrate, 20 μL of perovskite precursor was spin-coated and annealed 100 °C for 1 h. Then HTL in CB was spin-coated on the cooled samples. For the NVP system, the HTL/CB solution was dropped on perovskite film and followed by a spin-coating process at 3000 rpm for 30 s. TCSPC spectra were tested separately from the Spiro-OMeTAD side or SnO$_2$ side of the sample. For the preparation of cross-section samples, we pre-cut the FTO substrate with a glass cutter before film deposition and broke the sample after film deposition to obtain vertical, ordered cross-sectional samples.

**Focused-ion beam (FIB) for perovskite/NVP cross-sections for scanning transmission electron microscopy (STEM)**

The NVP-based perovskite was transferred to the chamber of the FEI Helios G4 CX dual-beam system and focused using the SEM model. 8 μm × 3 μm platinum was sputtered onto the perovskite/NVP sample to protect the internal morphology of the target slice. The sample was dug with a Ga$^+$ ion beam into two 4-μm deep holes adjacent to the target zone. A tungsten needle was welded to the target slice by sputtering platinum in between. The target zone was then inclined and cut off the bottom and side connections to sample substrate by Ga$^+$ ion beam. The tungsten needle transferred target slice to a TEM sample holder and welded together by sputtering platinum. Finally, the thickness of target slice was further reduced to 100 nm by using low-powered Ga$^+$ ion beam from the top platinum side, resulting in the ultra-thin target sample for HAADF-STEM.

**High-angle annular dark-field (HAADF) STEM**

The structures of the NVP-based perovskite were characterized using a FEI Talos F200X microscope in STEM mode at 200 kV, equipped with an EDS detector and a high-angle annular dark-field (HAADF) detector. Ultra-thin samples for STEM were prepared using FIB in a FEI Helios G4 CX dual beam microscope.

**Time-correlated single photon counting**

TCSPC spectra of perovskite films were excited with 520 nm laser (55 nJ cm$^{-2}$ fluence and 1 MHz repetition rate) impinged on either the HTL or ETL side.

**PSC fabrication for PCE measurements**

The perovskite solution was deposited on CBD-SnO$_2$/FTO by two consecutive spin-coating steps of 1000 rpm and 5000 rpm for 10 s and 30 s, respectively. During the second spin-coating step (5000 rpm), 100 μL of EA was quickly poured onto the substrate after 20 s. The films were then annealed at 100 °C for 1 h. The hole-transport layer was deposited by spin-coating a spiro-OMeTAD solution at 3000 rpm for 30 s. Finally, the gold electrode (80 nm) was deposited by thermal evaporation.

**Device characterization**

Current density-voltage ($J–V$) curves were obtained using a solar simulator (class 3 A, XES-40S3, SAN-EI) under AM 1.5 G standard light equipped with a Keithley 2400 source meter. The standard silicon solar cell (QE-B1) calibrated by NIM was used to calibrate the light intensity to AM 1.5 G irradiance (100 mW/cm$^2$). A shadow mask was used to define the effective aperture area of the device to 0.09 cm$^2$. PSCs with an 0.1 cm$^2$ active area were fabricated by evaporating top-electrodes with mask sizes of 0.32 cm × 0.32 cm. PSCs with a 1.0 cm$^2$ active area were fabricated by evaporating top-electrodes with mask sizes of 1.1 cm × 1.4 cm and then measured using a 1.0 cm × 1.0 cm aperture mask. All materials, solutions, and preparation processes for large-area perovskite films were the same as for the small-sized PSCs.

**Encapsulated PSC stability measurements**

PSCs were encapsulated by covering a glass slide (overlap size: 1.2 cm × 1.2 cm) on the device (substrate size: 1.4 cm × 1.4 cm, device area: 0.32 cm × 0.32 cm) with UV-curable resin (ThreeBond 3042B) for stability measurements. For ISOS-L-2 protocol, encapsulated PSCs were continuously illuminated under a solar simulator and heated on hot-plate at 65 °C in a N$_2$ glovebox. For the ISOS-D-3 protocol, encapsulated PSCs were stored in a damp-heat chamber (BPS-50CL, Shanghai BluePard, China) with a setting temperature of 85 °C and a relative humidity of 85%. PSCs were periodically measured (once per day except Sunday for 7 weeks) using a solar simulator (class 3 A, XES-40S3, SAN-EI).

**Unencapsulated PSC stability measurement and Water-vapor measurement**

The damp-heat (85 °C, 85% relative humidity) tests under dark conditions by unencapsulated PSCs fabricated with NVP as the only solvent device were performed using in the environment test chamber (BPS-50CL, Shanghai BluePard, China) for 720 min.

For water-vapor test, 500 mL of deionized water was added to a beaker and heated to boiling on a hot plate, and the unencapsulated control and perovskite/pure NVP films were fumigated in hot water vapor to observe the film changes.

**Lead leakage measurement**

To measure the lead leakage from the perovskite devices, the unencapsulated control and PCSs with 15% NVP, 30% NVP, 60% NVP, and 100% NVP were immersed in glass vials containing 10 mL of deionized water, respectively, and the devices were removed after 60 min. All samples in the glass vials were analyzed by flame atomic absorption spectrometer (iCETM 3500, Thermo Fisher).

**Reporting summary**

Further information on research design is available in the Nature Portfolio Reporting Summary linked to this article.

## Data availability

All data generated in this study are provided in the article and Supplementary Information and the raw data supporting this study are available from the Source Data file. Source data are provided with this paper.

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

## Acknowledgements

We thank the Analytical & Testing Center at Northwestern Polytechnical University for FIB-STEM measurement, SAXS/WAXS beamline at the Australian Synchrotron for GIWAXS measurement, Hong Kong Polytechnic University for the academic license of Silvaco software. This work is supported financially by the National Natural Science Foundation of China (62075094 T.Q., 52003118 F.W., 62205143 W.H.); Natural Science Foundation of Jiangsu Province (BK20211537 T.Q.).

## Author contributions

T.Q. and F.W. conceived the idea and designed the experiment; F.W., T.Q., and W.H. supervised the work; M.L., Q.T., R.S., and H.M. fabricated the perovskite devices and carried out the PV performance characterizations; F.W. provided STEM data analysis; M.L. and R.S. carried out the KPFM and PF-TUNA measurement; Q.T., H.W., and J.C. carried out the TCSPC, SCLC, and EIS measurement; Z.L. carried out the SEM measurement with the assistance of Z.W.; J.C. and H.C. provided architecture simulation; A.W. carried out the HR-TEM measurement with the assistance of J.Z.; J.D. provided GIWAXS data analysis with the assistance of Y.C.; Yo.L. and H.M. carried out built-in electric field testing; J.L. and Ya.L. performed the IR measurement; X.C., P.G., and S.Y. performed the XRD and XPS measurement; Y.S., T.L., and W.L. performed the Trap density measurement and PLQE measurement with the assistance of Q.T.; F.W. performed the data analysis and wrote the manuscript. T.Q. provided some revisions. R.L., J.W., Y-B.C., and X.L. gave some useful suggestions. All authors discussed the results and commented on the manuscript.

## Competing interests

The authors declare no competing interests.
