## [Peer Review File · Nature Communications]

Monolithically-Grained Perovskite Solar Cell with Mortise-Tenon Structure for Charge Extraction BalanceREVIEWER COMMENTS

Reviewer #1 (Remarks to the Author):

In this work, Wang et.al presents a very innovative strategy to address two important issues in the research of perovskite solar cells (PSCs), including disorder crystallization of perovskite and unbalanced interface charge extraction. In this perovskite system, a novel in-situ polymerizing agent is used as the template to form monolithic perovskite grain and a unique “Mortise-Tenon” (M-T) structure, which can suppress non-radiative recombination and balance interface charge extraction. The proposed M-T structure of the perovskite film is innovative, and the experimental data can fully support this conclusion. The M-T structured PSC achieves a remarkable certified efficiency of 24.55%, and good long-term stability according to ISOS-L-2 and ISOS-D-3 protocols. This work reports important advances for the field of highly efficient and stable PSCs. Therefore, I believe this work deserves being published in Nature Communications.

1.NVP as additive in the perovskite precursor convert into PVP via ATRP during annealing, and form M-T structure after spin-coating HTL solution. I am interested in whether the M-T structure can be formed if a polymer is used as the additive directly or using a small molecule additive that can't be polymerized. Therefore, I suggest using PVP and NMP as additives in perovskite films to investigate the morphology changes as the Figure 2c and 2d.

2.NVP acts as perovskite additive and eventually polymerize on the grain boundaries of the perovskite film. Does the addition affect the perovskite band-gap?

3.The author modulated the molar ratios of NVP addition in perovskite such as 15%, 30%, 60%, and 100%. And PSCs with 30 mol% exhibited the champion device performance. In Supplementary Fig. 8., an increasing passivation effect was observed in the XPS measurement as the amount of NVP added increases. Please explain what affects the final efficiency of the device with 60% and 100% addition.

4.Please describe the trap density measurement by SCLC in detail.

5.For STEM images of Figure 1c and 1d, the scale-bar is too small. Please check the font size of all figures to increase the readability of the article.

6.For Figure 4f, please mark the perovskite films as the control or NVP treated films independently to avoid confusion.

Reviewer #2 (Remarks to the Author):

In this work, authors described in-situ polymerization of NVP monomers allow guiding the formation of monolithic perovskite grains, wherein hole-transport layer (HTL) is diffused into to result in so called "Mortise-Tenon" (M-T) structure. Due to the improved contact with HTL and the passivation of PVP polymers, PSCs containing monolithic perovskite grains exhibited the mitigated non-radiative recombination and balanced interface charge extraction, hence yielding excellent PCEs and stabilities. As results, the PSCs achieved certified efficiency of 24.55% and maintain >95% initial efficiency over 1100 h illumination. Overall, this is a nice work addressing challenges of perovskite solar cells. Reviewer would like to recommend this work is suitable for the publication in Nature Communication.

Some technique comments for further improvement:

1. There lacks evidence on Figure 2a that perovskite is caged by NVPs in precursor solution. This is an important aspect to illustrate additives guiding the nucleation and crystal growth of perovskites. Therefore, characterization on the precursor solution is needed.
2. The description of Fig.1 b is absent in the figure caption; The description of Fig.2b is inaccurate. Perhaps it should be "b, Perovskite precursor is soluble in pure NVP and is miscible with DMSO (top);" The dash lines in Fig.3e are somehow misleading, as the length of 630 nm or 667 nm.
3. When estimating the defect concentration by SCLC method, the authors would need considering the influence of M-T structure. Since HTL penetrated into the perovskite layers, to use the height of perovskite grains as the average thickness might not accurate.
4. Wondering if PVP templates help mitigating the lead leakage of perovskite solar cells.

Reviewer #3 (Remarks to the Author):

The work titled "Monolithically-Grained Perovskite Solar Cell with Mortise-Tenon Structure for Charge Extraction Balance" by Wang et al. demonstrates an in-situ polymerized molecule NVP as the additive in perovskite film, which can form monolithic perovskite grain and a unique "Mortise-Tenon" (M-T) structure. The high-quality crystallization and M-T structure of perovskite films show several advantages, such as suppressed non-radiative recombination and balanced interface charge extraction. The underlined mechanism of the formation of M-T structure is demonstrated clearly and the experimental data strongly support the conclusion. This strategy provides a breakthrough in terms of concept, efficient performance, and high stability. An impressive certified PCE of 24.55% is obtained and the stability measurement according to the ISOS-protocols is convinced.

In summary, this paper shows impressive results and thorough discussion, I strongly recommend this work publication in Nature Communications. Some possible experiments for further improvement are listed as below.

1 In this study, the formation of the M-T structure of perovskite film is well confirmed by STEM, and the formation mechanism is demonstrated experimentally. The author proposes that a significant improvement in hole extraction is achieved, which is attributed to the enhanced hole extraction by the M-T structure and is proved by CAFM, TCSPC measurements and device simulations. However, in general additives can effectively change the interfacial energy level structure of perovskite films and thus affect carrier extraction. I suggest to test interfacial energy band structure of the perovskite films by ultraviolet photoelectron spectroscopy to further verify the origin of the enhancement of the hole extraction.

2 In Figure 2b, it is interesting to notice that NVP has an excellent solubility for the perovskite precursor. How about applying NVP as a solvent for fabricating perovskite films or even solar cells. As discussed by the author, the polymerized network may endow the device with good moisture stability.

3 The NVP-based PSCs exhibited impressive stability. In this discussion part, I was amazed by the water vapor tests on the unencapsulated perovskite/NVP film, however, the description of this experiment is brief. Please describe this experiment in detail, including the perovskite component and experimental conditions, and give an explanation about the super stability.

4 For evaluating the light and thermal stability, the author uses PTAA as the HTL instead of Spiro-OMeTAD. Please explain the reason and give the detailed description about the stability measurement in the experimental section.

Reply to reviewer #1

In this work, Wang et.al presents a very innovative strategy to address two important issues in the research of perovskite solar cells (PSCs), including disorder crystallization of perovskite and unbalanced interface charge extraction. In this perovskite system, a novel in-situ polymerizing agent is used as the template to form monolithic perovskite grain and a unique “Mortise-Tenon” (M-T) structure, which can suppress non-radiative recombination and balance interface charge extraction. The proposed M-T structure of the perovskite film is innovative, and the experimental data can fully support this conclusion. The M-T structured PSC achieves a remarkable certified efficiency of 24.55%, and good long-term stability according to ISOS-L-2 and ISOS-D-3 protocols. This work reports important advances for the field of highly efficient and stable PSCs. Therefore, I believe this work deserves being published in Nature Communications.

General Response: We appreciate the reviewer for reviewing our manuscript and providing constructive comments to improve the work. We carefully considered these comments and the detailed response can be found in the point-to-point response below.

Comment #1: NVP as additive in the perovskite precursor convert into PVP via ATRP during annealing, and form M-T structure after spin-coating HTL solution. I am interested in whether the M-T structure can be formed if a polymer is used as the additive directly or using a small molecule additive that can't be polymerized. Therefore, I suggest using PVP and NMP as additives in perovskite films to investigate the morphology changes as the Figure 2c and 2d.

Response: Thanks for the reviewer's important suggestion about the universal use of NVP. As the reviewer's suggestion, we have applied NMP and PVP as the perovskite additive to investigate the morphology changes of the perovskite film. And we found that both NMP and PVP could not form the same morphological structure as NVP after CB extraction. We have added the discussion in the revised manuscript and related characterization Supplementary Information as below.

Changes in the manuscript:

“Small molecular N-methyl-2-pyrrolidone (NMP) and PVP polymer were used as the additives in perovskite films. As shown in Supplementary Fig. 7, the unpolymerized NMP additives showed a similar morphology as the control perovskite film. In contrast, the PVP polymer additive showed very small and disordered crystallization of perovskite due to rapid precipitation during annealing. Both NMP and PVP could not form the same morphological structure as NVP after CB extraction.”

Changes in the Supplementary Information.

Supplementary Fig. 7 Surface SEM images of **a**, the perovskite/NMP films and **b**, perovskite/PVP film before and after CB extraction, the scale bar is 1 μm.

Comment #2: NVP acts as perovskite additive and eventually polymerize on the grain boundaries of the perovskite film. Does the addition affect the perovskite band-gap?

Response: Many thanks for the reviewer's suggestion. We measured the UV absorption of the perovskite film and confirmed that the perovskite band gap was not changed at 1.55eV after addition of NVP compared to the control film as shown in Supplementary Fig. 11 a.

Changes in the Supplementary Information.

Supplementary Fig.11. a, UV-vis absorption spectra of the control and perovskite/NVP film.

Comment #3: The author modulated the molar ratios of NVP addition in perovskite such as 15%, 30%, 60%, and 100%. And PSCs with 30 mol% exhibited the champion device performance. In Supplementary Fig. 8., an increasing passivation effect was observed in the XPS measurement as the amount of NVP added increases. Please explain what affects the final efficiency of the device with 60% and 100% addition.

Response: Thanks for the reviewer's important comment. In the XPS measurement, increasing passivation effects were observed due to the increase of passivated C=O groups in NVP. However, when the ratio of NVP increased to 60% and 100%, large amount of NVP addition probably affected the crystallization of perovskite, and thus decreased the device efficiency. We have added the discussion in the revised manuscript and related characterization Supplementary Information as below.

Changes in the manuscript:

Change 1: "SEM images in Supplementary Fig. 16 clearly showed that further increasing the proportion of NVP to 60% and 100%, large amount of NVP addition probably affected the crystallization of perovskite, and thus decreased the device efficiency."

Changes in the Supplementary Information.

Supplementary Fig.16. Surface SEM images of perovskite films with different molar ratios of NVP, the scale bar is 1μm.

Comment #4: Please describe the trap density measurement by SCLC in detail.

Response: We sincerely appreciate review's important comment.

Considering the Reviewer 2's comment on the SCLC method in this M-T structured perovskite system.

“Reviewer2-Comment #3: When estimating the defect concentration by SCLC method, the authors would need considering the influence of M-T structure. Since HTL penetrated into the perovskite layers, to use the height of perovskite grains as the average thickness might not accurate.” In order to eliminate the effects that might be caused by the thickness of perovskite, we estimated the defect concentration by time-correlated single photon counting (TCSPC) and fitted by a generic kinetic model according to the methodology of the references [Ref.46. *Nature* **562**, 249-253 (2018). Ref.46 *Phys. Rev. Appl.* 2, 034007 (2014)]. We also carried out the PL quantum efficiencies (PLQEs) to evaluate the quality of the perovskite film. We have added the discussion and the related characterization in the revised manuscript and Supplementary Information as below.

Changes in the manuscript

“Furthermore, as shown in Fig. 2f, perovskite/NVP film exhibited the longer perovskite lifetime (1138.42 ns) and lower trap density ($1.58 \times 10^{15} \text{ cm}^{-3}$) compared to the control film (375.69 ns and $8.11 \times 10^{15} \text{ cm}^{-3}$) measured by time-correlated single photon counting (TCSPC) and fitted by a generic kinetic model^{46,47}. Supplementary Fig. 9 showed that with increasing the excitation density, the PL quantum efficiencies (PLQEs) gradually reaching a maximum value owing to the filling of defects. Perovskite/NVP film showed a higher PLQE values with a maximum of 10.20% compared to the control film (8.38%). TCSPC and PLQE measurement confirmed that perovskite/NVP film with the monolithic grain and passivation effect synergistically suppressed the defect-related non-radiative recombination.”

Fig. 2. f, Time-resolved photoluminescence decay curves (excitation: 520 nm, 2.26 nJ cm^{-2} , 0.1MHz). Solid lines were fitted from the generic kinetic model to obtain the trap density of perovskite films.

Changes in the Supplementary Information:

Supplementary Fig. 9. Excitation-intensity-dependent PLQE of control perovskite and perovskite/NVP films.

Comment #5: For STEM images of Figure 1c and 1d, the scale-bar is too small. Please check the font size of all figures to increase the readability of the article.

Response: Thanks for the reviewer's suggestion. For increasing the readability of the article, we have checked and changed the font size of all the figures carefully in the revised manuscript.

Comment #6: For Figure 4f, please mark the perovskite films as the control or NVP treated films independently to avoid confusion.

Response: Thanks for the reviewer's suggestion. We have marked the picture of Figure 4f in the revised manuscript.

Reply to reviewer #2

In this work, authors described in-situ polymerization of NVP monomers allow guiding the formation of monolithic perovskite grains, wherein hole-transport layer (HTL) is diffused into to result in so called “Mortise-Tenon” (M-T) structure. Due to the improved contact with HTL and the passivation of PVP polymers, PSCs containing monolithic perovskite grains exhibited the mitigated non-radiative recombination and balanced interface charge extraction, hence yielding excellent PCEs and stabilities. As results, the PSCs achieved certified efficiency of 24.55% and maintain >95% initial efficiency over 1100 h illumination. Overall, this is a nice work addressing challenges of perovskite solar cells. Reviewer would like to recommend this work is suitable for the publication in Nature Communication.

General Response: We thank the reviewer for the nice comment and well appreciating the importance of our work. We carefully considered these comments and the detailed response can be found in the point-to-point response below.

Comment #1: There lacks evidence on Figure 2a that perovskite is caged by NVPs in precursor solution. This is an important aspect to illustrate additives guiding the nucleation and crystal growth of perovskites. Therefore, characterization on the precursor solution is needed.

Response: We sincerely appreciate the review's valuable comment and helpful suggestion. We absolutely agreed the review's opinion. This is an important question for our understanding of the interaction between NVP and perovskite precursors. In this regard, for investigating the interaction between the NVP and perovskite precursors, ^1H NMR, IR spectra and a solubility experiment were carried out in the revised manuscript. We have added the discussion in the revised manuscript and related characterization in Supplementary Information as below.

Changes in the manuscript

“After spin-coating process, NVP could still surround the perovskite crystal seeds at the initial nucleation step owing to strong interaction between NVP and perovskite precursors (PbI_2 and FA^+) proved by ^1H NMR, IR spectra and a solubility experiment as shown in Supplementary Fig. 5 and Supplementary Discussion 1.”

Changes in the Supplementary Information:

Supplementary Fig. 5. **a**, Photos of pure NVP, NVP/PbI₂ mixture, and NVP/PbI₂/FAI mixture. **b**, comparison of ^1H NMR spectra of NVP, NVP/PbI₂, NVP/FAI/ PbI₂, and FAI. **c**. FTIR spectra of the NVP, NVP/PbI₂, and PVP/PbI₂ (NVP/PbI₂ film annealing 100 °C for 1h).

Supplementary Discussion 1.

For directly investigate the interaction of NVP and perovskite precursors, as shown in Supplementary Fig. 5a, PbI_2 was added into the pure NVP, leading to a formation of yellow precipitation, then after adding FAI into this mixture, the precipitation disappeared and the solution became clarified. This observation suggested a synergistic interaction between NVP, FAI and PbI_2 . ^1H NMR spectra as shown in Supplementary Fig. 5b also proved this speculation. With the addition of FAI in NVP, the single resonance peak (8.788 ppm) of the ammonium in FAI split into two at 8.973 ppm and 8.608 ppm, implying that the strong interaction between FA^+ and NVP. The variation of ^1H NMR spectra of the NVP/ PbI_2 solution and NVP is minimal, indicating that the interaction between NVP and PbI_2 in solution is negligible. However, this changed substantially in the case of solid films. As shown in Supplementary Fig. 5c, Fourier Transform Infrared Spectroscopy (FTIR) confirmed that C=O of NVP could strongly coordinate with Pb^{2+} . For pure NVP, the stretching vibration of the C=O bond at 1704 cm^{-1} shifted to lower wavenumber at 1694 cm^{-1} after addition of PbI_2 , and further shifted to 1639 cm^{-1} after NVP annealing to PVP polymer. The apparent shift of the C=O absorption peak indicated the strong coordination between NVP and Pb^{2+} in the solid film state.

Comment #2: The description of Fig.1 b is absent in the figure caption; The description of Fig.2b is inaccurate. Perhaps it should be “b, Perovskite precursor is soluble in pure NVP and is miscible with DMSO (top);” The dash lines in Fig.3e are somehow misleading, as the length of 630 nm or 667 nm.

Response: Many thanks for the reviewer’s comment. We have checked the full manuscript carefully and all errors and misleading Figures have been corrected in the revised manuscript.

Comment #3: When estimating the defect concentration by SCLC method, the authors would need considering the influence of M-T structure. Since HTL penetrated into the perovskite layers, to use the height of perovskite grains as the average thickness might not accurate.

Response: We sincerely appreciate the reviewer’s important comment. In order to eliminate the effects that might be caused by the thickness of perovskite, we estimated the defect concentration by time-correlated single photon counting (TCSPC) and fitted by a generic kinetic model according to the methodology of the references. [Ref.46. *Nature* **562**, 249-253 (2018). Ref.46 *Phys. Rev. Appl.* **2**, 034007 (2014)]. We also carried out the PL quantum efficiencies (PLQEs) to evaluate the quality of the perovskite film. We have added the discussion and the related characterization in the revised manuscript and Supplementary Information as below.

Changes in the manuscript

“Furthermore, as shown in Fig. 2f, perovskite/NVP film exhibited the longer perovskite lifetime (1138.42 ns) and lower trap density ($1.58 \times 10^{15} \text{ cm}^{-3}$) compared to the control film (375.69 ns and $8.11 \times 10^{15} \text{ cm}^{-3}$) measured by time-correlated single photon counting (TCSPC) and fitted by a generic kinetic model^{46,47}. Supplementary Fig. 9 showed that with increasing the excitation density, the PL quantum efficiencies (PLQEs) gradually reaching a maximum value owing to the filling of defects. Perovskite/NVP film showed a higher PLQE values with a maximum of 10.20% compared to the control film (8.38%). TCSPC and PLQE measurement confirmed that perovskite/NVP film with the monolithic grain and passivation effect synergistically suppressed the defect-related non-radiative recombination.”

Fig. 2. f, Time-resolved photoluminescence decay curves (excitation: 520 nm, 2.26 nJ cm^{-2} , 0.1MHz). Solid lines were fitted from the generic kinetic model to obtain the trap density of perovskite films.

Changes in the Supplementary Information:

Supplementary Fig. 9. Excitation-intensity-dependent PLQE of control perovskite and perovskite/NVP films.

Comment #4: Wondering if PVP templates help mitigating the lead leakage of perovskite solar cells.

Response: We sincerely appreciate review's important suggestion. Actually, the perovskite with NVP addition indeed exhibited excellent temperature and humidity stability. As the reviewer's suggestion, we measured the lead leakage of unencapsulated PSCs with different ratio of NVP by using by flame atomic absorption spectrometer (AAS), and found that the NVP-based PSCs could mitigate the lead leakage efficiently. We have added the discussion in the revised manuscript and related characterization in Supplementary Information as below.

Changes in the manuscript

“To evaluate whether the polymer network would suppress lead leakage from the device, unencapsulated PSCs with different ratio of NVP were immersed into deionized water for 1 hour. The lead concentration of the solution was analyzed via flame atomic absorption spectrometer (AAS) as shown in Supplementary Fig. 22. The Pb concentration of the control PSC was 4.86 ppm, and as the molar ratio of NVP increased from 15% to 100%, the Pb concentration decreased to 1.85 ppm, which was one-fourth of that in the control counterpart.”

Changes in the Supplementary Information:

Supplementary Fig. 22. The lead ion concentration in the aqueous solution after perovskite films soaking in water with different molar ratios of NVP for 1 hour. The lead ion concentration in water was determined by atomic absorption spectrophotometer (AAS) measurement.

Lead leakage measurement: To measure the lead leakage from the perovskite devices, unencapsulated the control and PCSs with 15% NVP, 30% NVP, 60% NVP and 100% NVP were

immersed in glass vials containing 10 mL of deionized water, respectively, and the devices were removed after 60 min. All samples in the glass vials were analyzed by flame atomic absorption spectrometry (iCETM 3500, Thermo Fisher).

Reply to reviewer #3

The work titled "Monolithically-Grained Perovskite Solar Cell with Mortise-Tenon Structure for Charge Extraction Balance" by Wang et al. demonstrates an in-situ polymerized molecule NVP as the additive in perovskite film, which can form monolithic perovskite grain and a unique "Mortise-Tenon" (M-T) structure. The high-quality crystallization and M-T structure of perovskite films show several advantages, such as suppressed non-radiative recombination and balanced interface charge extraction. The underlined mechanism of the formation of M-T structure is demonstrated clearly and the experimental data strongly support the conclusion. This strategy provides a breakthrough in terms of concept, efficient performance, and high stability. An impressive certified PCE of 24.55% is obtained and the stability measurement according to the ISOS-protocols is convinced.

In summary, this paper shows impressive results and thorough discussion, I strongly recommend this work publication in Nature Communications. Some possible experiments for further improvement are listed as below.

General Response: We are grateful to the reviewer for the nice comments and well appreciating the importance of our work. We carefully considered these comments and the detailed response can be found in the point-to-point response below.

Comment #1: In this study, the formation of the M-T structure of perovskite film is well confirmed by STEM, and the formation mechanism is demonstrated experimentally. The author proposes that a significant improvement in hole extraction is achieved, which is attributed to the enhanced hole extraction by the M-T structure and is proved by CAFM, TCSPC measurements and device simulations. However, in general additives can effectively change the interfacial energy level structure of perovskite films and thus affect carrier extraction. I suggest to test interfacial energy band structure of the perovskite films by ultraviolet photoelectron spectroscopy to further verify the origin of the enhancement of the hole extraction.

Response: Thanks for the reviewer’s important suggestion. We absolutely agree the reviewer’s opinion. As suggested by the reviewer, we have added the related discussion in the revised manuscript and UPS measurement in Supplementary Information as below.

Changes in the manuscript: “To further verify the origin of the enhancement of the hole extraction, ultraviolet photoelectron spectroscopy (UPS) was carried out to detect the interfacial energy level structure of the perovskite films. As shown in Supplementary Fig.11, the control and perovskite/NVP films exhibited comparable valence band maximum. Therefore, the effective enhancement on hole extraction came mainly from the M-T structural contact between perovskite and HTL.”

Changes in the Supplementary Information:

Supplementary Fig. 11. **a,** UV-vis absorption spectra of the control and perovskite/NVP film. **b,** Valence-band region and photoemission cut off energy of the UPS spectra. **c,** Energy-level diagram constructed from UPS results. Conduction band minimum (E_c), Valence band maximum (E_v), and Fermi level (E_f).

Comment #2: In Figure 2b, it is interesting to notice that NVP has an excellent solubility for the perovskite precursor. How about applying NVP as a solvent for fabricating perovskite films or even solar cells. As discussed by the author, the polymerized network may endow the device with good moisture stability.

Response: We sincerely appreciate the reviewer’s important suggestion. Actually, during the optimization of perovskite solar cells (PSCs), we have optimized PSCs using pure NVP as solvent to fabricate perovskite films and done the stability tests as below.

Changes in the manuscript: “Furthermore, owing to the excellent solubility of NVP for perovskite precursor, unencapsulated PSCs used pure NVP as the only solvent to fabricate perovskite film. The device demonstrated PCE of 17.24% and maintained >80% of its initial PCE over 600 minutes, and the control PSC degraded immediately.”

Changes in the Supplementary Information:

Supplementary Fig. 23. a, J - V curves measured at reverse and forward scan of 1.474 M $(\text{FAPbI}_3)_{0.95}(\text{MAPbBr}_3)_{0.05}$ precursor was dissolved in 1 mL of pristine NVP solvent PSCs. **b,** Tracking of environmental stability of devices in an environment (85 °C, 85% RH) for 720 min and the images of unencapsulated devices.

Changes in the Supplementary Information:

“Preparation of the perovskite layer: For the pure NVP system, pure NVP was used as the only solvent to dissolve the perovskite precursors.”

“Unencapsulated PSC stability measurement: The damp-heat (85 °C, 85% relative humidity) tests under dark conditions by unencapsulated PSCs fabricated with NVP as the only solvent device were performed using in the environment test chamber (BPS-50CL, Shanghai BluePard, China) for 720 min.”

Comment #3: The NVP-based PSCs exhibited impressive stability. In this discussion part, I was amazed by the water vapor tests on the unencapsulated perovskite/NVP film, however, the description of this experiment is brief. Please describe this experiment in detail, including the perovskite component and experimental conditions, and give an explanation about the super stability.

Response: We appreciate to the reviewer's important question. We apologize for inadvertently missing this important information. Actually, owing to the excellent solubility of NVP for perovskite precursor, the NVP-based perovskite performed in Water vapor test was using the pure NVP as the only solvent to fabricate perovskite film. The experiment in detail was in the Supplementary Information. This superior stability might be attribute to the monolithic perovskite/NVP grains covered by polymerized network of PVP, which also confirmed by XPS measurement. The experiment of water-vapor test, including the perovskite component and experimental conditions was added in the experiment section of Supplementary Information.

Changes in the manuscript: “Water-vapor test provided a better visualization of the role of perovskite with NVP addition in retarding perovskite decomposition. We fumigated the unencapsulated control and perovskite/pure NVP films in hot water vapor and could see that the control films rapidly turned yellow while the NVP films remained in the black phase (Fig. 4f and Supplementary Movie 1).”

Changes in the Supplementary Information:

“Preparation of the perovskite layer: For the pure NVP system, pure NVP was used as the only solvent to dissolve the perovskite precursors.”

“Water-vapor measurement: For water vapor test, 500 ml of deionized water was added to a beaker and heated to boiling on a hot plate, and the unencapsulated control and perovskite/pure NVP films were fumigated in hot water vapor to observe the film changes.”

Comment #4: For evaluating the light and thermal stability, the author uses PTAA as the HTL instead of Spiro-OMeTAD. Please explain the reason and give the detailed description about the stability measurement in the experimental section.

Response: Thanks for the reviewer's important suggestion. Kim et al. found that PTAA is more stable than Spiro-OMeTAD under thermal stress at both 65 °C and 85 °C. [Sol. Energy Mater. Sol. Cells, 2017, 162, 41–46.] Due to the better thermal stability of PTAA, many reports used PTAA instead of spiro-OMeTAD as the hole transport layer for thermal testing. [Nat. Commun. 2023, 14, 839; Nat. Photonics, 2019, 13, 460; Science, 2021, 371, 1359; Nat. Commun. 2023, 14,1342; Angew. Chem. Int. Ed. 2022, e202211259]. We have added the related references and the detailed description about the stability measurement in the experimental section of Supplementary Information.

Changes in the experiment section of Supplementary Information:

“For the device thermal stability test, PTAA doped with 4-Isopropyl-4'-methyldiphenyliodonium Tetrakis (pentafluorophenyl) borate (TPFB) was used to replace spiro-OMeTAD as the hole transport layer. The concentration of PTAA was 30 mg mL⁻¹ and the weight ratio of PTAA/TPFB was 10:1. The PTAA was deposited on top of the perovskite layer at a spin rate of 3000 rpm. for 30s.”

REVIEWERS' COMMENTS

Reviewer #1 (Remarks to the Author):

This manuscript has been well modified and the authors responded all the comments properly. I am satisfied with that and the revision could be accepted in Nature Communications.

Reviewer #2 (Remarks to the Author):

The revised manuscript has addressed my concerns and can be published as it is.

Reviewer #3 (Remarks to the Author):

revision is ok.